# Borda Regret Minimization for Generalized Linear Dueling Bandits

## Abstract

Dueling bandits are widely used to model preferential feedback prevalent in many applications such as recommendation systems and ranking. In this paper, we study the Borda regret minimization problem for dueling bandits, which aims to identify the item with the highest Borda score while minimizing the cumulative regret. We propose a rich class of generalized linear dueling bandit models, which cover many existing models. We first prove a regret lower bound of order $\Omega(d^{2/3}T^{2/3})$ for the Borda regret minimization problem, where $d$ is the dimension of contextual vectors and $T$ is the time horizon. To attain this lower bound, we propose an explore-then-commit type algorithm for the stochastic setting, which has a nearly matching regret upper bound $\widetilde{O}(d^{2/3}T^{2/3})$. We also propose an EXP3-type algorithm for the adversarial setting, where the underlying model parameter can change at each round. Our algorithm achieves an $\widetilde{O}(d^{2/3}T^{2/3})$ regret, which is also optimal. Empirical evaluations on both synthetic data and a simulated real-world environment are conducted to corroborate our theoretical analysis.

## 1 Introduction

Multi-armed bandits (MAB) (Lattimore and Szepesvári, 2020) is an interactive game where at each round, an agent chooses an arm to pull and receives a noisy reward as feedback. In contrast to numerical feedback considered in classic MAB settings, preferential feedback is more natural in various online learning tasks including information retrieval Yue and Joachims (2009), recommendation systems Sui and Burdick (2014), ranking Minka et al. (2018), crowdsourcing Chen et al. (2013), etc. Moreover, numerical feedback is also more difficult to gauge and prone to errors in many real-world applications. For example, when provided with items to shop or movies to watch, it is more natural for a customer to pick a preferred one than scoring the options. This motivates *Dueling Bandits* (Yue and Joachims, 2009), where the agent repeatedly pulls two arms at a time and is provided with feedback being the binary outcome of "duels" between the two arms.

In dueling bandits problems, the outcome of duels is commonly modeled as Bernoulli random variables due to their binary nature. At each round, suppose the agent chooses to compare arm $i$ and $j$, then the binary feedback is assumed to be sampled independently from a Bernoulli distribution. For a dueling bandits instance with $K$ arms, the probabilistic model of the instance can be fully characterized by a $K \times K$ preference probability matrix with each entry being: $p_{i,j} = \mathbb{P}\left(\text{arm } i \text{ is chosen over arm } j\right)$.

In a broader range of applications such as ranking, "arms" are often referred to as "items". We will use these two terms interchangeably in the rest of this paper. One central goal of dueling bandits is to devise a strategy to identify the "optimal" item as quickly as possible, measured by either sample complexity or cumulative regret. However, the notion of optimality for dueling bandits is way harder to define than for multi-armed bandits. The latter can simply define the arm with the highest numerical feedback as the optimal arm, while for dueling bandits there is no obvious definition solely dependent on $\{p_{i,j} | i, j \in [K]\}$.

Submitted to 37th Conference on Neural Information Processing Systems (NeurIPS 2023). Do not distribute.

The first few works on dueling bandits imposed strong assumptions on $p_{i,j}$. For example, Yue et al. (2012) assumed that there exists a true ranking that is coherent among all items, and the preference probabilities must satisfy both strong stochastic transitivity (SST) and stochastic triangle inequality (STI). While relaxations like weak stochastic transitivity (Falahatgar et al., 2018) or relaxed stochastic transitivity (Yue and Joachims, 2011) exist, they typically still assume the true ranking exists and the preference probabilities are consistent, i.e., $p_{i,j} > \frac{1}{2}$ if and only if $i$ is ranked higher than $j$. In reality, the existence of such coherent ranking aligned with item preferences is rarely the case. For example, $p_{i,j}$ may be interpreted as the probability of one basketball team $i$ beating another team $j$, and there can be a circle among the match advantage relations.

In this paper, we do not assume such coherent ranking exists and solely rely on the *Borda score* based on preference probabilities. The Borda score $B(i)$ of an item $i$ is the probability that it is preferred when compared with another random item, namely $B(i) := \frac{1}{K-1} \sum_{j \neq i} p_{i,j}$. The item with the highest Borda score is called the *Borda winner*. The Borda winner is intuitively appealing and always well-defined for any set of preferential probabilities. The Borda score also does not require the problem instance to obey any consistency or transitivity, and it is considered one of the most general criteria.

To identify the Borda winner, estimations of the Borda scores are needed. Since estimating the Borda score for one item requires comparing it with every other items, the sample complexity is prohibitively high when there are numerous items. On the other hand, in many real-world applications, the agent has access to side information that can assist the evaluation of $p_{i,j}$. For instance, an e-commerce item carries its category as well as many other attributes, and the user might have a preference for a certain category (Wang et al., 2018). For a movie, the genre and the plot as well as the directors and actors can also be taken into consideration when making choices (Liu et al., 2017).

Based on the above motivation, we consider *Generalized Linear Dueling Bandits*. At each round, the agent selects two items from a finite set of items and receives a comparison result of the preferred item. The comparisons depend on known intrinsic contexts/features associated with each pair of items. The contexts can be obtained from upstream tasks, such as topic modeling (Zhu et al., 2012) or embedding (Vasile et al., 2016). Our goal is to adaptively select items and minimize the regret with respect to the optimal item (i.e., Borda winner). Our main contributions are summarized as follows:

- We show a hardness result regarding the Borda regret minimization for the (generalized) linear model. We prove a worst-case regret lower bound $\Omega(d^{2/3}T^{2/3})$ for our dueling bandit model, showing that even in the stochastic setting, minimizing the Borda regret is difficult. The construction and proof of the lower bound are new and might be of independent interest.
- We propose an explore-then-commit type algorithm under the stochastic setting, which can achieve a nearly matching upper bound $\widetilde{O}(d^{2/3}T^{2/3})$. When the number of items $K$ is small, the algorithm can also be configured to achieve a smaller regret $\widetilde{O}\big((d\log K)^{1/3}T^{2/3}\big)$.
- We propose an EXP3 type algorithm for linear dueling bandits under the adversarial setting, which can achieve a nearly matching upper bound $\widetilde{O}\big((d\log K)^{1/3}T^{2/3}\big)$.
- We conduct empirical studies to verify the correctness of our theoretical claims. Under both synthetic and real-world data settings, our algorithms can outperform all the baselines in terms of cumulative regret.

**Notation**  In this paper, we use normal letters to denote scalars, lowercase bold letters to denote vectors, and uppercase bold letters to denote matrices. For a vector $\mathbf{x}$, $\|\mathbf{x}\|$ denotes its $\ell_2$-norm. The weighted $\ell_2$-norm associated with a positive-definite matrix $\mathbf{A}$ is defined as $\|\mathbf{x}\|_{\mathbf{A}} = \sqrt{\mathbf{x}^\top \mathbf{A}\mathbf{x}}$. The minimum eigenvalue of a matrix $\mathbf{A}$ is written as $\lambda_{\min}(\mathbf{A})$. We use standard asymptotic notations including $O(\cdot), \Omega(\cdot), \Theta(\cdot)$, and $\widetilde{O}(\cdot), \widetilde{\Omega}(\cdot), \widetilde{\Theta}(\cdot)$ will hide logarithmic factors. For a positive integer $N$, $[N] := \{1, 2, \ldots, N\}$.

## 2  Related Work

**Multi-armed and Contextual Bandits**  Multi-armed bandit is a problem of identifying the best choice in a sequential decision-making system. It has been studied in numerous ways with a wide range of applications (Even-Dar et al., 2002; Lai et al., 1985; Kuleshov and Precup, 2014). Contextual linear bandit is a special type of bandit problem where the agent is provided with side information, i.e., contexts, and rewards are assumed to have a linear structure. Various algorithms (Rusmevichientong

and Tsitsiklis, 2010; Filippi et al., 2010; Abbasi-Yadkori et al., 2011; Li et al., 2017; Jun et al., 2017) have been proposed to utilize this contextual information.

**Dueling Bandits and Its Performance Metrics** Dueling bandits is a variant of MAB with preferential feedback (Yue et al., 2012; Zoghi et al., 2014a, 2015). A comprehensive survey can be found at Bengs et al. (2021). As discussed previously, the probabilistic structure of a dueling bandits problem is governed by the preference probabilities, over which an optimal item needs to be defined. Optimality under the *Borda score* criteria has been adopted by several previous works (Jamieson et al., 2015; Falahatgar et al., 2017a; Heckel et al., 2018; Saha et al., 2021a). The most relevant work to ours is Saha et al. (2021a), where they studied the problem of regret minimization for adversarial dueling bandits and proved a $T$-round Borda regret upper bound $\widetilde{O}(K^{1/3}T^{2/3})$. They also provide an $\Omega(K^{1/3}T^{2/3})$ lower bound for stationary dueling bandits using Borda regret.

Apart from the Borda score, *Copeland score* is also a widely used criteria (Urvoy et al., 2013; Zoghi et al., 2015, 2014b; Wu and Liu, 2016; Komiyama et al., 2016). It is defined as $C(i) := \frac{1}{K-1}\sum_{j\neq i} \mathbb{1}\{p_{i,j} > 1/2\}$. A Copeland winner is the item that beats the most number of other items. It can be viewed as a "thresholded" version of Borda winner. In addition to Borda and Copeland winners, optimality notions such as a von Neumann winner were also studied in Ramamohan et al. (2016); Dudík et al. (2015); Balsubramani et al. (2016).

Another line of work focuses on identifying the optimal item or the total ranking, assuming the preference probabilities are consistent. Common consistency conditions include Strong Stochastic Transitivity (Yue et al., 2012; Falahatgar et al., 2017a,b), Weak Stochastic Transitivity (Falahatgar et al., 2018; Ren et al., 2019; Wu et al., 2022; Lou et al., 2022), Relaxed Stochastic Transitivity (Yue and Joachims, 2011) and Stochastic Triangle Inequality. Sometimes the aforementioned transitivity can also be implied by some structured models like the Bradley–Terry model. We emphasize that these consistency conditions are not assumed or implicitly implied in our setting.

**Contextual Dueling Bandits** In Dudík et al. (2015), contextual information is incorporated in the dueling bandits framework. Later, Saha (2021) studied a structured contextual dueling bandits setting where each item $i$ has its own contextual vector $\mathbf{x}_i$ (sometimes called Linear Stochastic Transitivity). Each item then has an intrinsic score $v_i$ equal to the linear product of an unknown parameter vector $\boldsymbol{\theta}^*$ and its contextual vector $\mathbf{x}_i$. The preference probability between two items $i$ and $j$ is assumed to be $\mu(v_i - v_j)$ where $\mu(\cdot)$ is the logistic function. These intrinsic scores of items naturally define a ranking over items. The regret is also computed as the gap between the scores of pulled items and the best item. While in this paper, we assume that the contextual vectors are associated with item pairs and define regret on the Borda score. In Section A.1, we provide a more detailed discussion showing that the setting considered in Saha (2021) can be viewed as a special case of our model.

# 3 Backgrounds and Preliminaries

## 3.1 Problem Setting

We first consider the stochastic preferential feedback model with $K$ items in the fixed time horizon setting. We denote the item set by $[K]$ and let $T$ be the total number of rounds. At each round $t$, the agent can pick any pair of items $(i_t, j_t)$ to compare and receive stochastic feedback about whether item $i_t$ is preferred over item $j_t$, (denoted by $i_t \succ j_t$). We denote the probability of seeing the event $i \succ j$ as $p_{i,j} \in [0,1]$. Naturally, we assume $p_{i,j} + p_{j,i} = 1$, and $p_{i,i} = 1/2$.

In this paper, we are concerned with the generalized linear model (GLM), where there is assumed to exist an *unknown* parameter $\boldsymbol{\theta}^* \in \mathbb{R}^d$, and each pair of items $(i,j)$ has its own *known* contextual/feature vector $\boldsymbol{\phi}_{i,j} \in \mathbb{R}^d$ with $\|\boldsymbol{\phi}_{i,j}\| \leq 1$. There is also a fixed known link function (sometimes called comparison function) $\mu(\cdot)$ that is monotonically increasing and satisfies $\mu(x) + \mu(-x) = 1$, e.g. a linear function or the logistic function $\mu(x) = 1/(1 + e^{-x})$. The preference probability is defined as $p_{i,j} = \mu(\boldsymbol{\phi}_{i,j}^\top \boldsymbol{\theta}^*)$. At each round, denote $r_t = \mathbb{1}\{i_t \succ j_t\}$, then we have

$$\mathbb{E}[r_t|i_t, j_t] = p_{i_t,j_t} = \mu(\boldsymbol{\phi}_{i_t,j_t}^\top \boldsymbol{\theta}^*).$$

Then our model can also be written as

$$r_t = \mu(\boldsymbol{\phi}_{i_t,j_t}^\top \boldsymbol{\theta}^*) + \epsilon_t,$$

where the noises $\{\epsilon_t\}_{t\in[T]}$ are zero-mean, 1-sub-Gaussian and assumed independent from each other. Note that, given the constraint $p_{i,j} + p_{j,i} = 1$, it is implied that $\phi_{i,j} = -\phi_{j,i}$ for any $i \in [K], j \in [K]$.

The agent's goal is to maximize the cumulative Borda score. The (slightly modified [1]) Borda score of item $i$ is defined as $B(i) = \frac{1}{K}\sum_{j=1}^{K} p_{i,j}$, and the Borda winner is defined as $i^* = \operatorname{argmax}_{i\in[K]} B(i)$. The problem of merely identifying the Borda winner was deemed trivial (Zoghi et al., 2014a; Busa-Fekete et al., 2018) because for a fixed item $i$, uniformly random sampling $j$ and receiving feedback $r_{i,j} = \text{Bernoulli}(p_{i,j})$ yield a Bernoulli random variable with its expectation being the Borda score $B(i)$. This so-called *Borda reduction* trick makes identifying the Borda winner as easy as the best-arm identification for $K$-armed bandits. Moreover, if the regret is defined as $\text{Regret}(T) = \sum_{t=1}^{T} (B(i^*) - B(i_t))$, then any optimal algorithms for multi-arm bandits can achieve $\widetilde{O}(\sqrt{T})$ regret.

However, the above definition of regret does not respect the fact that a pair of items are selected at each round. When the agent chooses two items to compare, it is natural to define the regret so that both items contribute equally. A commonly used regret, e.g., in Saha et al. (2021a), has the following form:

$$\text{Regret}(T) = \sum_{t=1}^{T} \big(2B(i^*) - B(i_t) - B(j_t)\big), \tag{1}$$

where the regret is defined as the sum of the sub-optimality of both selected arms. Sub-optimality is measured by the gap between the Borda scores of the compared items and the Borda winner. This form of regret deems any classical multi-arm bandit algorithm with Borda reduction vacuous because taking $j_t$ into consideration will invoke $\Theta(T)$ regret.

**Adversarial Setting** Saha et al. (2021b) considered an adversarial setting for the multi-armed case, where at each round $t$, the comparison follows a potentially different probability model, denoted by $\{p_{i,j}^t\}_{i,j\in[K]}$. In this paper, we consider its contextual counterpart. Formally, we assume there is an underlying parameter $\boldsymbol{\theta}_t^*$, and at round $t$, the preference probability is defined as $p_{i,j}^t = \mu(\boldsymbol{\phi}_{i,j}^\top \boldsymbol{\theta}_t^*)$.

The Borda score of item $i \in [K]$ at round $t$ is defined as $B_t(i) = \frac{1}{K}\sum_{j=1}^{K} p_{i,j}^t$, and the Borda winner at round $T$ is defined as $i^* = \operatorname{argmax}_{i\in[K]} \sum_{t=1}^{T} B_t(i)$. The $T$-round regret is thus defined as $\text{Regret}(T) = \sum_{t=1}^{T} \big(2B_t(i^*) - B_t(i_t) - B_t(j_t)\big)$.

## 3.2 Assumptions

In this section, we present the assumptions required for establishing theoretical guarantees. Due to the fact that the analysis technique is largely extracted from Li et al. (2017), we follow them to make assumptions to enable regret minimization for generalized linear dueling bandits.

We make a regularity assumption about the distribution of the contextual vectors:

**Assumption 1.** There exists a constant $\lambda_0 > 0$ such that $\lambda_{\min}\big(\frac{1}{K^2}\sum_{i=1}^{K}\sum_{j=1}^{K} \boldsymbol{\phi}_{i,j}\boldsymbol{\phi}_{i,j}^\top\big) \geq \lambda_0$.

This assumption is only utilized to initialize the design matrix $\mathbf{V}_\tau = \sum_{t=1}^{\tau} \boldsymbol{\phi}_{i_t,j_t}\boldsymbol{\phi}_{i_t,j_t}^\top$ so that the minimum eigenvalue is large enough. We follow Li et al. (2017) to deem $\lambda_0$ as a constant.

We also need the following assumption regarding the link function $\mu(\cdot)$:

**Assumption 2.** Let $\dot{\mu}$ be the first-order derivative of $\mu$. We have $\kappa := \inf_{\|\mathbf{x}\|\leq 1, \|\boldsymbol{\theta}-\boldsymbol{\theta}^*\|\leq 1} \dot{\mu}(\mathbf{x}^\top\boldsymbol{\theta}) > 0$.

Assuming $\kappa > 0$ is necessary to ensure the maximum log-likelihood estimator can converge to the true parameter $\boldsymbol{\theta}^*$ (Li et al., 2017, Section 3). This type of assumption is commonly made in previous works for generalized linear models (Filippi et al., 2010; Li et al., 2017; Faury et al., 2020).

Another common assumption is regarding the continuity and smoothness of the link function.

---

[1]Previous works define Borda score as $B_i' = \frac{1}{K-1}\sum_{j\neq i} p_{i,j}$, excluding the diagonal term $p_{i,i} = 1/2$. Our definition is equivalent since the difference between two items satisfies $B(i) - B_j = \frac{K-1}{K}(B_i' - B_j')$. Therefore, the regret will be in the same order for both definitions.

**Assumption 3.** $\mu$ is twice differentiable. Its first and second-order derivatives are upper-bounded by constants $L_\mu$ and $M_\mu$ respectively.

This is a very mild assumption. For example, it is easy to verify that the logistic link function satisfies Assumption 3 with $L_\mu = M_\mu = 1/4$.

# 4  The Hardness Result

$$
\text{"good"} \left\{ \begin{array}{c} \\ \\ \\ \end{array} \right.
\text{"bad"} \left\{ \begin{array}{c} \\ \\ \end{array} \right.
\left[
\begin{array}{ccc|ccc}
\frac{1}{2} & \cdots & \frac{1}{2} & & & \\
\vdots & \ddots & \vdots & \multicolumn{3}{c}{\frac{3}{4}+} \\
\frac{1}{2} & \cdots & \frac{1}{2} & \multicolumn{3}{c}{\langle\phi_{i,j},\boldsymbol{\theta}\rangle} \\
\hline
& & & \frac{1}{2} & \cdots & \frac{1}{2} \\
\multicolumn{3}{c|}{\frac{1}{4}+} & \vdots & \ddots & \vdots \\
\multicolumn{3}{c|}{\langle\phi_{j,i},\boldsymbol{\theta}\rangle} & \frac{1}{2} & \cdots & \frac{1}{2}
\end{array}
\right]
$$

$$
\begin{array}{ccc}
\frac{3}{4}+\langle\mathbf{bit}(0),\boldsymbol{\theta}\rangle & \frac{3}{4}+\langle\mathbf{bit}(0),\boldsymbol{\theta}\rangle & \cdots \frac{3}{4}+\langle\mathbf{bit}(0),\boldsymbol{\theta}\rangle \\
\frac{3}{4}+\langle\mathbf{bit}(1),\boldsymbol{\theta}\rangle & \frac{3}{4}+\langle\mathbf{bit}(1),\boldsymbol{\theta}\rangle & \cdots \frac{3}{4}+\langle\mathbf{bit}(1),\boldsymbol{\theta}\rangle \\
\vdots & \vdots & \ddots \vdots \\
\frac{3}{4}+\langle\mathbf{bit}(2^d-1),\boldsymbol{\theta}\rangle & \frac{3}{4}+\langle\mathbf{bit}(2^d-1),\boldsymbol{\theta}\rangle & \cdots \frac{3}{4}+\langle\mathbf{bit}(2^d-1),\boldsymbol{\theta}\rangle
\end{array}
$$

Figure 1: Illustration of the hard-to-learn preference probability matrix $\{p_{i,j}^{\boldsymbol{\theta}}\}_{i\in[K],j\in[K]}$. There are $K = 2^{d+1}$ items in total. The first $2^d$ items are "good" items with higher Borda scores, and the last $2^d$ items are "bad" items. The upper right block $\{p_{i,j}\}_{i<2^d,j\geq 2^d}$ is defined as shown in the blue bubble. The lower left block satisfies $p_{i,j} = 1 - p_{j,i}$. For any $\boldsymbol{\theta}$, there exist one and only best item $i$ such that $\mathbf{bit}(i) = \mathbf{sign}(\boldsymbol{\theta})$.

This section presents Theorem 4, a worst-case regret lower bound for the stochastic linear dueling bandits. The proof of Theorem 4 relies on a class of hard instances, as shown in Figure 1. We show that any algorithm will incur a certain amount of regret when applied to this hard instance class. The constructed hard instances follow a stochastic linear model, which is a sub-class of the generalized linear model. Saha et al. (2021b) first proposed a similar construction for finite many arms with no contexts. Our construction is for a contextual setting and the proof of the lower bound takes a rather different route.

For any $d > 0$, we construct the class of hard instances as follows. An instance is specified by a vector $\boldsymbol{\theta} \in \{-\Delta, +\Delta\}^d$. The instance contains $2^{d+1}$ items (indexed from 0 to $2^{d+1} - 1$). The preference probability for an instance is defined by $p_{i,j}^{\boldsymbol{\theta}}$ as:

$$
p_{i,j}^{\boldsymbol{\theta}} = \begin{cases} \frac{1}{2}, & \text{if } i < 2^d, j < 2^d \text{ or if } i \geq 2^d, j \geq 2^d \\ \frac{3}{4}, & \text{if } i < 2^d, j \geq 2^d \\ \frac{1}{4}, & \text{if } i \geq 2^d, j < 2^d \end{cases} + \langle\phi_{i,j},\boldsymbol{\theta}\rangle,
$$

and the $d$-dimensional feature vectors $\phi_{i,j}$ are given by

$$
\phi_{i,j} = \begin{cases} \mathbf{0}, & \text{if } i < 2^d, j < 2^d \text{ or if } i \geq 2^d, j \geq 2^d \\ \mathbf{bit}(i), & \text{if } i < 2^d, j \geq 2^d \\ -\mathbf{bit}(j), & \text{if } i \geq 2^d, j < 2^d, \end{cases}
$$

where $\mathbf{bit}(\cdot)$ is the (shifted) bit representation of non-negative integers, i.e., suppose $x$ has the binary representation $x = b_0 \times 2^0 + b_1 \times 2^1 + \cdots + b_{d-1} \times 2^{d-1}$, then

$$
\mathbf{bit}(x) = (2b_0 - 1, 2b_1 - 1, \ldots, 2b_{d-1} - 1) = 2\boldsymbol{b} - 1.
$$

Note that $\mathbf{bit}(\cdot) \in \{-1, +1\}^d$, and that $\phi_{i,j} = -\phi_{j,i}$ is satisfied. The definition of $p_{i,j}^{\boldsymbol{\theta}}$ can be slightly tweaked to fit exactly the model described in Section 3 (see Remark 11 in Appendix).

Some calculation shows that the Borda scores of the $2^{d+1}$ items are:

$$
B^{\boldsymbol{\theta}}(i) = \begin{cases} \frac{5}{8} + \frac{1}{2}\langle\mathbf{bit}(i),\boldsymbol{\theta}\rangle, & \text{if } i < 2^d, \\ \frac{3}{8}, & \text{if } i \geq 2^d. \end{cases}
$$

Intuitively, the former half of items (those indexed from 0 to $2^d - 1$) are "good" items (one among them is optimal, others are nearly optimal), while the latter half of items are "bad" items. Under such hard instances, every time one of the two pulled items is a "bad" item, then a one-step regret

$B^{\boldsymbol{\theta}}(i^*) - B^{\boldsymbol{\theta}}(i) \geq 1/4$ is incurred. To minimize regret, we should thus try to avoid pulling "bad" items. However, in order to identify the best item among all "good" items, comparisons between "good" and "bad" items are necessary. The reason is simply that comparisons between "good" items give no information about the Borda scores as the comparison probabilities are $p_{i,j}^{\boldsymbol{\theta}} = \frac{1}{2}$ for all $i, j < 2^d$. Hence, any algorithm that can decently distinguish among the "good" items has to pull "bad" ones for a fair amount of times, and large regret is thus incurred. A similar observation is also made by Saha et al. (2021a).

This specific construction emphasizes the intrinsic hardness of Borda regret minimization: to differentiate the best item from its close competitors, the algorithm must query the bad items to gain information.

Formally, this class of hard instances leads to the following regret lower bound for both stochastic and adversarial settings:

**Theorem 4.** For any algorithm $\mathcal{A}$, there exists a hard instance $\{p_{i,j}^{\boldsymbol{\theta}}\}$ with $T > 4d^2$, such that $\mathcal{A}$ will incur expected regret at least $\Omega(d^{2/3}T^{2/3})$.

The construction of this hard instance for linear dueling bandits is inspired by the worst-case lower bound for the stochastic linear bandit (Dani et al., 2008), which has the order $\Omega(d\sqrt{T})$, while ours is $\Omega(d^{2/3}T^{2/3})$. The difference is that for the linear or multi-armed stochastic bandit, eliminating bad arms can make further exploration less expensive. But in our case, any amount of exploration will not reduce the cost of further exploration. This essentially means that exploration and exploitation must be separate, which is also supported by the fact that a simple explore-then-commit algorithm shown in Section 5 can be nearly optimal.

# 5 Stochastic Contextual Dueling Bandit

## 5.1 Algorithm Description

---
**Algorithm 1** BETC-GLM

---
1: **Input:** time horizon $T$, number of items $K$, feature dimension $d$, feature vectors $\boldsymbol{\phi}_{i,j}$ for $i \in [K]$, $j \in [K]$, exploration rounds $\tau$, error tolerance $\epsilon$, failure probability $\delta$.
2: **for** $t = 1, 2, \ldots, \tau$ **do**
3:     sample $i_t \sim \text{Uniform}([K])$, $j_t \sim \text{Uniform}([K])$
4:     query pair $(i_t, j_t)$ and receive feedback $r_t$
5: **end for**
6: Find the G-optimal design $\pi(i, j)$ based on $\boldsymbol{\phi}_{i,j}$ for $i \in [K], j \in [K]$
7: Let $N(i,j) = \left\lceil \frac{d\pi(i,j)}{\epsilon^2} \right\rceil$ for any $(i,j) \in \text{supp}(\pi)$, denote $N = \sum_{i=1}^{K} \sum_{j=1}^{K} N(i,j)$
8: **for** $i \in [K], j \in [K], s \in [N(i,j)]$ **do**
9:     set $t \leftarrow t + 1$, set $(i_t, j_t) = (i, j)$
10:     query pair $(i_t, j_t)$ and receive feedback $r_t$
11: **end for**
12: Calculate the empirical MLE estimator $\widehat{\boldsymbol{\theta}}_{\tau+N}$ based on all $\tau + N$ samples via (2)
13: Estimate the Borda score for each item:

$$\widehat{B}(i) = \frac{1}{K}\sum_{j=1}^{K}\mu(\boldsymbol{\phi}_{i,j}^{\top}\widehat{\boldsymbol{\theta}}_{\tau+N}), \qquad \widehat{i} = \underset{i \in [K]}{\text{argmax}}\ \widehat{B}(i)$$

14: Keep querying $(\widehat{i}, \widehat{i})$ for the rest of the time.

---

We propose an algorithm named Borda Explore-Then-Commit for Generalized Linear Models (BETC-GLM), presented in Algorithm 1. Our algorithm is inspired by the algorithm for generalized linear models proposed by Li et al. (2017).

At the high level, Algorithm 1 can be divided into two phases: the exploration phase (Line 2-11) and the exploitation phase (Line 12-14). The exploration phase ensures that the MLE estimator $\widehat{\boldsymbol{\theta}}$ is accurate enough so that the estimated Borda score is within $\widetilde{O}(\epsilon)$-range of the true Borda score

(ignoring other quantities). Then the exploitation phase simply chooses the empirical Borda winner to incur small regret.

During the exploration phase, the algorithm first performs "pure exploration" (Line 2-5), which can be seen as an initialization step for the algorithm. The purpose of this step is to ensure the design matrix $\mathbf{V}_{\tau+N} = \sum_{t=1}^{\tau+N} \phi_{i_t,j_t} \phi_{i_t,j_t}^\top$ is positive definite.

After that, the algorithm will perform the "designed exploration". Line 6 will find the G-optimal design, which minimizes the objective function $g(\pi) = \max_{i,j} \|\phi_{i,j}\|_{\mathbf{V}(\pi)^{-1}}^2$, where $\mathbf{V}(\pi) := \sum_{i,j} \pi(i,j)\phi_{i,j}\phi_{i,j}^\top$. The G-optimal design $\pi^*(\cdot)$ satisfies $\|\phi_{i,j}\|_{\mathbf{V}(\pi^*)^{-1}}^2 \leq d$, and can be efficiently approximated by the Frank-Wolfe algorithm (See Remark 8 for a detailed discussion). Then the algorithm will follow $\pi(\cdot)$ found at Line 6 to determine how many samples (Line 7) are needed. At Line 8-11, there are in total $N = \sum_{i=1}^{K} \sum_{j=1}^{K} N(i,j)$ samples queried, and the algorithm shall index them by $t = \tau+1, \tau+2, \ldots, \tau+N$.

At Line 12, the algorithm collects all the $\tau+N$ samples and performs the maximum likelihood estimation (MLE). For the generalized linear model, the MLE estimator $\widehat{\boldsymbol{\theta}}_{\tau+N}$ satisfies:

$$\sum_{t=1}^{\tau+N} \mu(\phi_{i_t,j_t}^\top \widehat{\boldsymbol{\theta}}_{\tau+N})\phi_{i_t,j_t} = \sum_{t=1}^{\tau+N} r_t \phi_{i_t,j_t}, \tag{2}$$

or equivalently, it can be determined by solving a strongly concave optimization problem:

$$\widehat{\boldsymbol{\theta}}_{\tau+N} \in \operatorname*{argmax}_{\boldsymbol{\theta}} \sum_{t=1}^{\tau+N} \left( r_t \phi_{i_t,j_t}^\top \boldsymbol{\theta} - m(\phi_{i_t,j_t}^\top \boldsymbol{\theta}) \right),$$

where $\dot{m}(\cdot) = \mu(\cdot)$. For the logistic link function, $m(x) = \log(1 + e^x)$. As a special case of our generalized linear model, the linear model has a closed-form solution for (2). For example, if $\mu(x) = \frac{1}{2} + x$, i.e. $p_{i,j} = \frac{1}{2} + \phi_{i,j}^\top \boldsymbol{\theta}^*$, then (2) becomes:

$$\widehat{\boldsymbol{\theta}}_{\tau+N} = \mathbf{V}_{\tau+N}^{-1} \sum_{t=1}^{\tau+N} (r_t - 1/2)\phi_{i_t,j_t},$$

where $\mathbf{V}_{\tau+N} = \sum_{t=1}^{\tau+N} \phi_{i_t,j_t} \phi_{i_t,j_t}^\top$.

After the MLE estimator is obtained, Line 13 will calculate the estimated Borda score $\widehat{B}(i)$ for each item based on $\widehat{\boldsymbol{\theta}}_{\tau+N}$, and pick the empirically best one.

## 5.2 A Matching Regret Upper Bound

Algorithm 1 can be configured to tightly match the worst-case lower bound. The configuration and performance are described as follows:

**Theorem 5.** Suppose Assumption 1-3 hold and $T = \Omega(d^2)$. For any $\delta > 0$, if we set $\tau = C_4 \lambda_0^{-2}(d + \log(1/\delta))$ ($C_4$ is a universal constant) and $\epsilon = d^{1/6}T^{-1/3}$, then with probability at least $1 - 2\delta$, Algorithm 1 will incur regret bounded by:

$$O\left(\kappa^{-1} d^{2/3} T^{2/3} \sqrt{\log\left(T/d\delta\right)}\right).$$

By setting $\delta = T^{-1}$, the expected regret is bounded as $\widetilde{O}(\kappa^{-1} d^{2/3} T^{2/3})$.

For linear bandit models, such as the hard-to-learn instances in Section 4, $\kappa$ is a universal constant. Therefore, Theorem 5 tightly matches the lower bound in Theorem 4, up to logarithmic factors.

**Remark 6** (Regret for Fewer Arms). In typical scenarios, the number of items $K$ is not exponentially large in the dimension $d$. In this case, we can choose a different parameter set of $\tau$ and $\epsilon$ such that Algorithm 1 can achieve a smaller regret bound $\widetilde{O}(\kappa^{-1}(d\log K)^{1/3} T^{2/3})$ with smaller dependence on the dimension $d$. See Theorem 10 in Appendix A.2.

**Remark 7** (Regret for Infinitely Many Arms). In most practical scenarios of dueling bandits, it is adequate to consider a finite number $K$ of items (e.g., ranking items). Nonetheless, BETC-GLM

can be easily adapted to accommodate infinitely many arms in terms of regret. We can construct a covering over all $\phi_{i,j}$ and perform optimal design and exploration on the covering set. The resulting regret will be the same as our upper bound, i.e., $\widetilde{O}(d^{2/3}T^{2/3})$ up to some error caused by the epsilon net argument.

**Remark 8** (Approximate G-optimal Design). Algorithm 1 assumes an exact G-optimal design $\pi$ is obtained. In the experiments, we use the Frank-Wolfe algorithm to solve the constraint optimization problem (See Algorithm 5, Appendix G.3). To find a policy $\pi$ such that $g(\pi) \leq (1 + \varepsilon)g(\pi^*)$, roughly $O(d/\varepsilon)$ optimization steps are needed. Such a near-optimal design will introduce a factor of $(1 + \varepsilon)^{1/3}$ into the upper bounds.

# 6 Adversarial Contextual Dueling Bandit

This section addresses Borda regret minimization under the adversarial setting. As we introduced in Section 3.1, the unknown parameter $\boldsymbol{\theta}_t$ can vary for each round $t$, while the contextual vectors $\phi_{i,j}$ are fixed.

Our proposed algorithm, BEXP3, is designed for the contextual linear model. Formally, at round $t$ and given pair $(i, j)$, we have $p_{i,j}^t = \frac{1}{2} + \langle \phi_{i,j}, \boldsymbol{\theta}_t^* \rangle$.

## 6.1 Algorithm Description

---
**Algorithm 2** BEXP3

---
1: **Input:** time horizon $T$, number of items $K$, feature dimension $d$, feature vectors $\phi_{i,j}$ for $i \in [K]$, $j \in [K]$, learning rate $\eta$, exploration parameter $\gamma$.
2: **Initialize:** $q_1(i) = \frac{1}{K}$.
3: **for** $t = 1, \ldots, T$ **do**
4:     Sample items $i_t \sim q_t$, $j_t \sim q_t$.
5:     Query pair $(i_t, j_t)$ and receive feedback $r_t$
6:     Calculate $Q_t = \sum_{i \in [K]} \sum_{j \in [K]} q_t(i)q_t(j)\phi_{i,j}\phi_{i,j}^\top$, $\widehat{\boldsymbol{\theta}}_t = Q_t^{-1}\phi_{i_t,j_t}r_t$.
7:     Calculate the (shifted) Borda score estimates $\widehat{B}_t(i) = \langle \frac{1}{K}\sum_{j \in [K]} \phi_{i,j}, \widehat{\boldsymbol{\theta}}_t \rangle$.
8:     Update for all $i \in [K]$, set

$$\widetilde{q}_{t+1}(i) = \frac{\exp(\eta \sum_{l=1}^t \widehat{B}_l(i))}{\sum_{j \in [K]} \exp(\eta \sum_{l=1}^t \widehat{B}_l(j))}; \qquad q_{t+1}(i) = (1 - \gamma)\widetilde{q}_{t+1}(i) + \frac{\gamma}{K}.$$

9: **end for**

---

Algorithm 2 is adapted from the DEXP3 algorithm in Saha et al. (2021b), which deals with the adversarial multi-armed dueling bandit. Algorithm 2 maintains a distribution $q_t(\cdot)$ over $[K]$, initialized as uniform distribution (Line 2). At every round $t$, two items are chosen following $q_t$ independently. Then Line 6 calculates the one-sample unbiased estimate $\widehat{\boldsymbol{\theta}}_t$ of the true underlying parameter $\boldsymbol{\theta}_t^*$. Line 7 further calculates the unbiased estimate of the (shifted) Borda score. Note that the true Borda score at round $t$ satisfies $B_t(i) = \frac{1}{2} + \langle \frac{1}{K}\sum_{j \in [K]} \phi_{i,j}, \boldsymbol{\theta}_t^* \rangle$. $\widehat{B}_t$ instead only estimates the second term of the Borda score. This is a choice to simplify the proof. The cumulative estimated score $\sum_{l=1}^t \widehat{B}_l(i)$ can be seen as the estimated cumulative reward of item $i$ at round $t$. In Line 8, $q_{t+1}$ is defined by the classic exponential weight update, along with a uniform exploration policy controlled by $\gamma$.

## 6.2 Upper Bounds

Algorithm 2 can also be configured to tightly match the worst-case lower bound:

**Theorem 9.** Suppose Assumption 1 holds. If we set $\eta = (\log K)^{2/3}d^{-1/3}T^{-2/3}$ and $\gamma = \sqrt{\eta d/\lambda_0} = (\log K)^{1/3}d^{1/3}T^{-1/3}\lambda_0^{-1/2}$, then the expected regret is upper-bounded by

$$O\big((d \log K)^{1/3}T^{2/3}\big).$$

Note that the lower bound construction is for the linear model and has $K = O(2^d)$, thus exactly matching the upper bound.

## 7 Experiments

This section compares the proposed algorithm BETC-GLM with existing ones that are capable of minimizing Borda regret. We use random responses (generated from fixed preferential matrices) to interact with all tested algorithms. Each algorithm is run for 50 times over a time horizon of $T = 10^6$. We report both the mean and the standard deviation of the cumulative Borda regret and supply some analysis. The following list summarizes all methods we studies in this section, a more complete description of the methods and parameters are available in Appendix E: BETC-GLM(-MATCH): Algorithm 1 proposed in this paper with different parameters. UCB-BORDA: The UCB algorithm (Auer et al.,

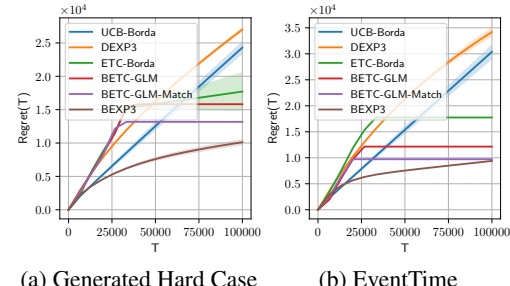

(a) Generated Hard Case  (b) EventTime

Figure 2: The regret of the proposed algorithms (BETC-GLM, BEXP3) and the baseline algorithms (UCB-BORDA, DEXP3, ETC-BORDA).

2002) using *Borda reduction*. DEXP3: Dueling-Exp3 developed by Saha et al. (2021a). ETC-BORDA: A simple explore-then-commit algorithm that does not take any contextual information into account. BEXP3: The proposed method for adversarial Borda bandits displayed in Algorithm 2.

**Generated Hard Case**  We first test the algorithms on the hard instances constructed in Section 4. We generate $\boldsymbol{\theta}^*$ randomly from $\{-\Delta, +\Delta\}^d$ with $\Delta = \frac{1}{4d}$ so that the comparison probabilities $p_{i,j}^{\boldsymbol{\theta}^*} \in [0,1]$ for all $i, j \in [K]$. We pick the dimension $d = 6$ and the number of arms is therefore $K = 2^{d+1} = 128$. Note the dual usage of $d$ in our construction and the model setup in Section 3.1. We refer readers to Remark 11 in Appendix B for more details.

As depicted in Figure 2a, the proposed algorithms (BETC-GLM, BEXP3) outperform the baseline algorithms in terms of cumulative regret when reaching the end of time horizon $T$. For UCB-BORDA, since it is not tailored for the dueling regret definition, it suffers from a linear regret as its second arm is always sampled uniformly at random, leading to a constant regret per round. DEXP3 and ETC-BORDA are two algorithms designed for $K$-armed dueling bandits. Both are unable to utilize contextual information and thus demand more exploration. As expected, their regrets are higher than BETC-GLM or BEXP3.

**Real-world Dataset**  To showcase the performance of the algorithms in a real-world setting, we use EventTime dataset (Zhang et al., 2016). In this dataset, $K = 100$ historical events are compared in a pairwise fashion by crowd-sourced workers. We first calculate the empirical preference probabilities $\widetilde{p}_{i,j}$ from the collected responses, and construct a generalized linear model based on the empirical preference probabilities. The algorithms are tested under this generalized linear model. Due to space limitations, more details are deferred to Appendix F.

As depicted in Figure 2b, the proposed algorithm BETC-GLM outperforms the baseline algorithms in terms of cumulative regret when reaching the end of time horizon $T$. The other proposed algorithm BEXP3 performs equally well even when misspecified (the algorithm is designed for linear setting, while the comparison probability follows a logistic model).

## 8 Conclusion and Future Work

In this paper, we introduced Borda regret into the generalized linear dueling bandits setting, along with an explore-then-commit type algorithm BETC-GLM and an EXP3 type algorithm BEXP3. The algorithms can achieve a nearly optimal regret upper bound, which we corroborate with a matching lower bound. The theoretical performance of the algorithms is verified empirically. It demonstrates superior performance compared to other baseline methods.

For future works, due to the fact that our exploration scheme guarantees an accurate estimate in all directions, our work can be extended to solve the top-k recovery or ranking problem, as long as a proper notion of regret can be identified.

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
