# OpenReview forum: "Borda Regret Minimization for Generalized Linear Dueling Bandits"
_NeurIPS.cc/2023/Conference — Submitted to NeurIPS 2023_

### Official Review · Reviewer_ECQz · 2023-06-18

**Soundness:** 3 good
**Presentation:** 3 good
**Contribution:** 2 fair
**Rating:** 6
**Confidence:** 3

**Summary:**

The authors consider the linear dueling bandit problem, where the regret is measured in terms of Borda regret. The Borda regret function was adopted as that used in Saha et al. 2021a. Different from conventional bandit settings, there is a mismatch between the reward function and the regret function definition in the sense that the expected reward of pulling a pair of arms is p_i,j, while the regret is defined based on the function B(i)+B(j), where B(i) is the Borda score of i-th arm. The authors provide a lower bound on the regret by the construction of hard cases and a matching algorithm based on explore-then-commit. The adversarial setting is also considered, where the EXP3 style algorithm is proposed and analyzed, and shown to match the upper bound.

**Strengths:**

+ For both settings, the proposed algorithms are shown to match the upper bounds.
+ The construction of hard cases is interesting.


**Weaknesses:**

- The experiment results are not very convincing.
- The algorithms are relatively straightforward extensions of the existing approaches
- The constructed hard case is also a relatively straightforward extension of that in Saha et al.


**Questions:**

- The setup for experiments is confusing. The two algorithms are for single context and adversarial settings, respectively. It is not clear why they can be used in the same setting for evaluation.

line 8 of algorithm 1: "s" should be "t"

---

> ### Author Rebuttal · Authors · 2023-08-08
>
> Thank you for your feedback. We address your comments and questions as follows:
>
> ---
>
> **Q1**:  The experiment results are not very convincing. The setup for experiments is confusing. The two algorithms are for single context and adversarial settings, respectively. It is not clear why they can be used in the same setting for evaluation. \
> **A1**: We are sorry if the description of the experiments is not clear enough. Here we will explain the two experimental settings respectively: \
> + The synthetic setting is in the linear stationary setting. Since BEXP3 can work for the linear adversarial setting (which admits the linear stationary setting as a special case), it also works for our linear stationary setting. We conducted this experiment as a sanity check for our theory. We would also like to mention that our hard instance construction (which the experiment is based on) can apply to both the stationary and the adversarial setting.
> + The real-world dataset is a non-linear stationary setting. BETC can work under this setting. BEXP3 is not designed for non-linear link function, but we still test whether it can work even under the misspecified situation, as explained in Line 338.
>
> Based on your feedback, we conducted an additional experiment to examine the performance of BEXP3. In this experiment, the number of items is $K = 64$, and the feature dimension is $d = 5$. The environment is adversarial and will alter its parameter $\theta^*_t$ (defined in Sec. 3) every 100 steps to make the algorithm’s chosen arms have the worst Borda score, introducing the largest one-step Borda regret. More specifically, we set $\theta^*\_t = \arg \min\_{\theta} B\_{\theta}(i_t)+B\_{\theta}(j_t)$. The whole simulation takes 100,000 steps, and we report the average cumulative regret over 100 runs with shaded areas near each line indicating the standard deviation. From the figure we uploaded, we can see that a non-adversarial algorithm (BETC-GLM) quickly suffers from a linear regret in the commit phase because the adversarial environment makes the committed arm the worst. DEXP3 and BEXP3 both can adapt to the adversarial environment, but our algorithm BEXP3 can also take advantage of the linear features, and thus incurs a smaller regret than DEXP3. Please see the figure we provided in the uploaded PDF file.
>
>
> ---
>
> **Q2**: The algorithms are relatively straightforward extensions of the existing approaches \
> **A2**:  Since the proposed setting has never been studied before, our main goal is to propose efficient algorithms that can optimally solve the new problem, rather than propose completely different algorithms from any previous one. In addition, it suffices to adopt some standard techniques such as explore-then-commit (ETC) or EXP3 in our algorithm design. Building upon standard techniques makes the algorithm easy to understand, implement, and analyze, which we consider as an advantage rather than a disadvantage.
>
> ---
>
> **Q3**: The constructed hard case is also a relatively straightforward extension of that in Saha et al. \
> **A3**:  We agree that our construction shares a similar high-level structure as in [Saha et al 21]. However, since the construction and the proof are for the contextual setting, it is quite different from the multi-armed setting. \
> In detail, our construction is based on the hardness of identifying the best arm in the $d$-dimensional linear bandit setting, which is different from [Saha et al 21].
> Besides, we used a different proof technique.  [Saha et al 21]’s proof is directly based on hypothesis testing: either identifying the best arm with gap $\epsilon$ within $T$ rounds (if $T > \frac{K}{1440 \epsilon^3}$) or incurring $\epsilon T$ regret (if $T \le \frac{K}{1440 \epsilon^3}$). In contrast, our proof technique bounds from below the regret by the expected number of sub-optimal arm pulls, and does not divide the problem instances into two cases (whether $T≤\frac{K}{1440 \epsilon^3}$).

---

> > ### Comment · Reviewer_ECQz · 2023-08-15
> >
> > Thanks for the explanation and additional experiment. The additional experiment helps. For the other two points, my feeling is still mixed. Yes, indeed simpler algorithms and bounds are good, but if the techniques are too similar to known ones, there is a concern about whether the new problem is too incremental. I am keeping the rating in this case.

---

### Official Review · Reviewer_43D4 · 2023-07-02

**Soundness:** 3 good
**Presentation:** 4 excellent
**Contribution:** 3 good
**Rating:** 6
**Confidence:** 4

**Summary:**

This paper studies the problem of minimising Borda regret for dueling bandits in the generalised linear setting where each pair of arms has a context vector associated with it.

The paper considers both the stochastic setting where the parameter $\theta^*$ used for generating rewards is fixed and the adversarial setting where it is time dependent.

**[Lower bound]** The authors show a hard problem instance for which any algorithm will incur a regret of $\Omega(d^{2/3}T^{2/3})$ in the stochastic setting, $d$ being the dimension of the context vectors.

**[BETC-GLM for stochastic setting]** The authors propose an explore-then-commit style algorithm that first pulls arm pairs based on a G-optimal design for some rounds and then exploits by selecting the arm $\hat{i}$ with the highest MLE Borda score and pulling $(\hat{i}, \hat{i})$. This algorithm incurs $\tilde{O}(\kappa^{-1} d^{2/3} T^{2/3})$ regret, where $\kappa$ is a problem dependent parameter (which is constant for linear bandits).

**[BEXP3]** The authors then propose an EXP style algorithm for the adversarial case. The algorithm is restricted to the linear setting. At each step, it estimates $\hat{\theta}_t$ using the reward obtained in that step, and uses it to estimate the Borda scores of all arms at that step. These scores are used to compute a distribution over arm pairs in the same spirit at EXP. This algorithm incurs a regret of $O((d \log K)^{1/3} T^{2/3})$, which matches the lower bound when $K = O(2^d)$.

Experiments on real and synthetic data corroborate the theoretical findings.



**Strengths:**

Originality - The Borda regret minimisation problem for generalised linear dueling bandits is new. The authors seem to have adequately cited the related work.

Quality - I have not checked the details in the appendix but the arguments presented in the main paper are inspired from well-known techniques, and appear sound to me. Barring a few issues listed in the weakness section, I believe that the claims are well supported, both theoretically and empirically via experiments on real and synthetic data.

Clarity - The paper is very well written and easy to understand. In particular, I appreciate the explanation for ``Borda reduction'' and why it is not sufficient in Section 3.1.

Significance - The results fill an important research gap.

**Weaknesses:**

Originality - The algorithms and their analysis have limited novelty. While I understand why Borda reduction does not trivially work, it would be helpful to have a summary of the technical challenges encountered in analysing the algorithms, while highlighting the new ideas that were employed.

Quality - BEXP3 assumes a linear model for $p_{i,j}^t$ and not a generalised linear model. I think this should be clarified in the abstract. It would also be useful to have a discussion on what makes having a link function here hard.

Clarity - Just two minor comments,
1. Use \citep instead of \citet wherever appropriate (e.g., L19-20)
2. Explaining the utility of using a G-optimal design in Algorithm 1 (to have a uniformly good estimate of $\theta$) will improve the readability of the paper.

Significance - The authors note in their conclusion that their exploration scheme guarantees accurate estimation in all directions, thereby paving way for extensions like top-k recovery and ranking problems. However, the exploration scheme is the G-optimal design, which is not a new contribution. Outside of this, the work seems to have limited impact.

**Questions:**

Please address the points under originality and significance in the weakness section.

**Limitations:**

I suggest adding some discussion addressing my concern under "Quality" in the weakness section as a limitation of the analysis for BEXP3 (if applicable, I may be wrong in which case please correct me).

---

> ### Author Rebuttal · Authors · 2023-08-08
>
> Thank you for your feedback. We appreciate that you recognize our work as “filling an important research gap”.  We address your comments and questions as follows:
>
> ---
>
> **Q1**:   The algorithms and their analysis have limited novelty. While I understand why Borda reduction does not trivially work, it would be helpful to have a summary of the technical challenges encountered in analyzing the algorithms while highlighting the new ideas that were employed. \
> **A1**:  Thank you for your suggestion. We summarize the challenges and new ideas below.
>
> + *Challenge (stochastic setting)*: Identify an effective policy to choose the pairs for comparison. \
> *Solution*: To estimate the Borda score, the most sample-efficient way is to query each pair uniformly. Under the contextual linear setting, this means we need to explore each direction in $\mathbb{R}^d$ uniformly well, leading to the G-optimal design-based exploration. Meanwhile,  our hard instance construction illustrates that exploration won’t help exploitation (Line 218-224). Therefore, it won’t hurt to separate exploration and exploitation. This observation leads to our simple yet efficient explore-then-commit algorithm.
>
> + *Challenge (adversarial setting)*: When applying the EXP3 framework, we have to bound the change in policy $\tilde{q}\_{t}$ at each round (Line 679), which requires an upper bound on $\hat{B}\_t(i)$. \
> *Solution*: To address this, we show in Lemma 17 that the Borda score is related to the inverse matrix norm $\max\_{i,j} \\| \phi\_{i,j} \\|^2\_{Q\_t}$ and further bound it by the minimum eigenvalue $\lambda\_0$ of the uniform matrix as in Assumption 1.
>
> + *Challenge (hard instance)*: Construct the hard instance for arbitrary $d$ dimension, and prove the lower bound. \
> *Solution*: The construction is based on the hardness of identifying the best arm in the $d$-dimensional linear bandit setting. To prove the lower bound, we first apply a reduction step to restrict the choice of $i_t$. Then we bound from below the regret by the expected number of sub-optimal arm pulls. The proof idea is new compared with previous works.
>
> ---
>
> **Q2**: BEXP3 assumes a linear model for p_{i,j} and not a generalized linear model. I think this should be clarified in the abstract. It would also be useful to have a discussion on what makes having a link function here hard. \
> **A2**: Thank you for your suggestion. We will make it clear that BEXP3 only works for linear models in the abstract. \
> Similar to the reason why EXP3-type algorithms for the adversarial generalized linear bandit are missing, the main difficulty of adding a link function to our adversarial setting is that, typically, EXP3 requires an unbiased one-sample estimator for the parameter (Line 6). For non-linear link functions, the widely used maximum-likelihood estimator is not unbiased, leading to the estimated Borda score being inaccurate. The bias cannot be sufficiently bounded with only one sample each round, and thus the regret becomes uncontrollable. This is the main difficulty why EXP3 cannot work well for generalized linear models, both for standard bandits and dueling bandits. We will add this to the discussion of the limitations of our algorithm.
>
> ---
>
> **Q3**: …However, the exploration scheme is the G-optimal design, which is not a new contribution. Outside of this, the work seems to have limited impact.\
> **A3**: We do not intend to claim the G-optimal design as our contribution. As we explained in A1, our contributions include the new problem setting, the hard instance, the algorithms, and the upper bounds. \
> Besides, one of the key takeaways from our paper is that Borda regret minimization is intrinsically hard, no matter if the environment is stationary or adversarial. Centered around this key observation, we derived the lower bound and two matching upper bounds. We believe this work is noteworthy to the research community especially when learning from human feedback (e.g., preferential feedback) has received increasing attention these days.

---

> > ### Comment · Reviewer_43D4 · 2023-08-14
> > **Thank you for the rebuttal**
> >
> > Thank you for taking time to respond to my questions.
> >
> > I had concerns about the novelty of this paper. In particular, the reward structure allows the authors to heavily borrow techniques from the generalised linear bandits (or adversarial linear bandits literature). For example, I considered the idea of needing a good estimate of $\theta$ in all directions, and using a G-optimal design to get it, to be fairly natural.
> >
> > Perhaps I was blinded by hindsight. Thank you for highlighting the challenges again. I have increased my score to 6. I hope that the authors will incorporate other suggestions from my comments in the next version of the manuscript. All the best and have a good day :)

---

> > > ### Author Response · Authors · 2023-08-14
> > > **Thank you!**
> > >
> > > Thank you again for your feedback on our paper. We really appreciate your increased score, and we're committed to incorporating your suggestions into the revised version.

---

### Official Review · Reviewer_LMZ8 · 2023-07-05

**Soundness:** 4 excellent
**Presentation:** 4 excellent
**Contribution:** 3 good
**Rating:** 7
**Confidence:** 3

**Summary:**

This paper discusses Borda regret minimization in a contextual dueling bandits scenario, where the context is given in a generalized linear form.
It provides a worst-case lower bound for the stochastic and adversarial learning scenario. The authors develop for both scenarios algorithmic solutions whose asymptotic Borda regret matches these lower bounds (up to logarithmic terms). These solutions are shown to empirically outperform current state-of-the art solutions in experiments.

**Strengths:**

The paper tackles an interesting problem. It contributes a novel lower bound for the particular learning scenario and develops algorithmic solutions, which (a) come with log-optimal regret upper bounds and (b) outperform current methods on both synthetic and real-world data.
In my opinion, this paper is well-written, the notation is convenient and the algorithms and proofs seem to be presented in a reader-friendly way.

**Weaknesses:**

I did not find any weaknesses while skimming the paper and the proofs.

While reading, I've collected some the following typos/minor suggestions:
- 170f.: $\lambda_{min}$ has not formally been introduced.
- 216: Refer here to the appendix for the proof of Thm. 4. Or did you mention before that proofs are to be found in the appendix?
- 240f.: Here, you call the G-optimal design $\pi^\ast$. For consistency, denote it also like this in Alg. 1 and the further discussion?
- 297: Refer to Thm.4 for convenience?
- 309: "we study"
- 331: [the] EventTime dataset
- 338: for [the/a] linear setting
- 457: be satisfied
- 522f.: \P_{\theta,\A} has not been defined
- 522f.: C=10?
- 529f.: Should averaging be done over $\theta \in \{-\Delta,\Delta\}$? Also, in 532ff. etc.
- 532f.: Why is "sign" in bold?
- 552f.: Regarding your identity of $V_\tau$ here, you seem to have switched from column- to rows for $\theta_{i,j}$ in comparison to l. 171f.
- 568f.: For readability, formally introduce $\succeq$
- 582: $|\mathrm{supp}(\pi)|$

**Questions:**

Does the hard instance in Remark 11 fulfill all of assumptions 1-3? (If trivial, at least mention it briefly for convenience.) If not, could you provide another hard instance fulfilling these, which leads to a similar bound?

**Limitations:**

The authors adequately addressed the limitations of their work.

---

> ### Author Rebuttal · Authors · 2023-08-08
>
> Thank you for your positive feedback. We address your comments and questions as follows:
>
> ---
>
> **Q1**:  While reading, I've collected the following typos/minor suggestions: … \
> **A1**: Thank you so much for the suggestions! We will fix them accordingly.
>
> ---
>
> **Q2**: Does the hard instance in Remark 11 fulfill all of assumptions 1-3? (If trivial, at least mention it briefly for convenience.) If not, could you provide another hard instance fulfilling these, which leads to a similar bound? \
> **A2**: The hard instance in Remark 11 satisfies all of Assumptions 1-3. The linear link function $F(x) = \frac{1+x}{2}$ automatically satisfies Assumptions 2 and 3. For Assumption 1, we have $\lambda_0 = 1/4$ so Assumption 1 holds too.

---

> > ### Comment · Reviewer_LMZ8 · 2023-08-15
> >
> > Thanks for your response, I keep the score.

---

### Official Review · Reviewer_7FMh · 2023-07-06

**Soundness:** 4 excellent
**Presentation:** 4 excellent
**Contribution:** 3 good
**Rating:** 7
**Confidence:** 3

**Summary:**


In dueling bandits, each pair of arms corresponds to some unknown probability $p_{i,j}$ where $p_{i,j}$ is the probability arm i is ranked higher than $j$. The learner sequentially chooses pairs of arms and receives a noisy result as to the ordering of the pair, i.e. Bernoulli$(p_{i,j})$. This paper considers dueling bandits under a generalised linear model where each pair of arms $i,j$ corresponds to a known context vector. The probability that $i$ is ranked higher $j$ is then the product of this context vector with some unknown vector $\theta^*$, passed through a link function. The goal of the learner is to pull arms with high ranking as much as possible, the regret on a single pair of arms is given as the average Borda score between the arms, and the learner aims to minimise cumulative regret. This setting has been considered previously, however, under the assumption that a hidden coherent ordering is forced on the arms. The key contribution of this paper is to relax this assumption and have essentially no constraint on the $p_{i,j}$s. In addition to the above setting, the authors consider a adversarial setting where the unknown vector $\theta^*$ is allowed to change round by round. Adversarial dueling bandits has been studied in the literature, the main contribution of this section is to extend this analysis to the contextual setting.

**Strengths:**

The paper is well written and easy to read. Sufficient treatment is given to previous works and care is taken to illustrate how the results of this paper are novel in comparison. In section 4 the class of hard instances is clearly constructed and the following discussion gives good intuition as to why explore then commit algorithms can work in this setting, which is in itself an interesting phenomenon. The results give a complete treatment of the setting, with matching upper and lower bounds, up to log terms. In their experiments the authors consider a variety of benchmarks, with application to synthetic and real world data.

**Weaknesses:**

I do not see why $\delta$ is given as input to the algorithm, it is not taken as a parameter but rather passed to the exploration phase $\tau$.

The discussion in section A.1 of the appendix is vital to understanding the novelty of this work in comparison to Saha 2021, it is a shame that this section is not present in the main text.



**Questions:**


Have the authors considered relaxed assumptions, that do not require a coherent ordering of the arms, but ensure that explore then commit algorithms cannot be optimal? For instance, a arm cannot be ranked higher than another arm whose Borda score is sufficiently higher than its own, according to some tolerance.

**Limitations:**

For the BETC-GLM algorithm, the authors acknowledge the potential limitation of not having access to the exact G optimal design. They suggest a well known sub routine to estimate the G optimal design and describe the additional error term that would incurred by this. The authors discuss and provide compelling arguments as to whether the assumptions are reasonable.

---

> ### Author Rebuttal · Authors · 2023-08-08
>
> Thank you for your positive feedback and strong support. We address your comments and questions as follows:
>
> ---
>
> **Q1**:  I do not see why $\delta$ is given as input to the algorithm, it is not taken as a parameter but rather passed to the exploration phase $\tau$. \
> **A1**: Thanks for pointing this out. Indeed, $\delta$ can be removed from the input of Algorithm 1 as it only appears in $\tau$. We will fix this.
>
> ---
>
> **Q2**: The discussion in Section A.1 of the appendix is vital to understanding the novelty of this work in comparison to Saha 2021, it is a shame that this section is not present in the main text. \
> **A2**: Thank you for the suggestion. Due to the space limit, we chose to leave this section in the appendix. We will re-arrange the content to fit section A.1 in the main text as per your suggestion.
>
> ---
>
> **Q3**: Have the authors considered relaxed assumptions, that do not require a coherent ordering of the arms, but ensure that explore then commit algorithms cannot be optimal? For instance, an arm cannot be ranked higher than another arm whose Borda score is sufficiently higher than its own, according to some tolerance. \
> **A3**: We would like to first clarify that our work does not assume a coherent ordering/ranking of the arms. We rephrase your question as follows: If there are more structures in the problem but still there is no coherent ranking, will the ETC algorithm become not optimal?   \
>   ***Our answer***: In general, we are not sure. But for the example you mentioned: if $B(i) - B(j) > \Delta$ where $\Delta$ is certain tolerance, then $i$ is always preferred over $j$ (i.e., $p_{ij} = 1$), it appears that a slightly modified ETC algorithm can still be optimal. Here is the argument: under your proposed assumption, if some arm has a low Borda score, its probability against all high-Borda-score arms is always $0$, so stopping exploring it early won’t change the Borda score gap among those high-Borda-score arms. Essentially, the regret is determined by those $\Delta$-near-optimal arms, instead of all arms. On the other hand, within the $\Delta$ radius,  our lower bound construction will still hold by rescaling $\langle \phi_{i,j}, \theta^* \rangle$. Therefore, the order of the lower bound won’t change and thus ETC can still be optimal.

---

> > ### Comment · Reviewer_7FMh · 2023-08-17
> >
> > Thank you for the detailed answer to my question, I agree, finding such a setting where an ETC approach is not optimal is not obvious. Instead of a fixed tolerance $\Delta$, perhaps something like the constraint $p_{i,j} \geq exp(-\alpha\Delta_{i,j})$ where $\Delta_{i,j}$ is the gap between Borda scores of $i$ and $j$ and $\alpha>0$?
> >
> > After reading the other reviews I am inclined to agree that the contribution of this work is limited by the lack of novelty in the algorithms, however, providing the lower bound and showing an explore then commit approach can be optimal in this setting, is a nice result in itself. I have slightly lowered my score but still recommend the paper to be accepted.

---

### Author Rebuttal · Authors · 2023-08-08

Dear reviewers,

Based on the feedback of Reviewer ECQz, we conducted an additional experiment to examine the performance of BEXP3. Please find the figure and description we provided in the uploaded PDF file.

In this experiment, the number of items is $K = 64$, and the feature dimension is $d = 5$. The environment is adversarial and will alter its parameter $\theta^*_t$ (defined in Sec. 3) every 100 steps to make the algorithm’s chosen arms have the worst Borda score, introducing the largest one-step Borda regret. More specifically, we set $\theta^*\_t = \arg \min\_{\theta} B\_{\theta}(i_t)+B\_{\theta}(j_t)$. The whole simulation takes 100,000 steps, and we report the average cumulative regret over 100 runs with shaded areas near each line indicating the standard deviation. From the figure we uploaded, we can see that a non-adversarial algorithm (BETC-GLM) quickly suffers from a linear regret in the commit phase because the adversarial environment makes the committed arm the worst. DEXP3 and BEXP3 both can adapt to the adversarial environment, but our algorithm BEXP3 can also take advantage of the linear features, and thus incurs a smaller regret than DEXP3.

---

### Decision · Program_Chairs · 2023-09-21

**Decision:**

Reject

**Comment:**

After considering the rebuttal, reviewers' comments, and looking at the paper more thoroughly, I tend to agree with the concerns raised by Reviewer ECQz and 43D4: The ideas used in the paper lack originality as the techniques and lower bounds claims presented in this paper are mostly aggregated from the literature of linear bandits (G-Optimal design), GLM bandits, and Dueling bandits for bora scores. The primary contribution/ significance of the paper is in question due to the lack of originality in their problem formulation, algorithmic and mathematical novelty, especially given the NeurIPS bar.

Factoring these concerns, we decided not to proceed with an acceptance this time, however, we urge the authors to incorporate all the reviewers' suggestions and please resubmit the work to the next suitable venue (possibly incorporating some results that could potentially contribute a new idea over the existing techniques).